# All-optical steering on the proton emission in laser-induced nanoplasmas

Fenghao Sun[1,2], Qiwen Qu[1], Hui Li [1] ✉, Shicheng Jiang [1] ✉, Qingcao Liu[3], Shuai Ben[4], Yu Pei[4], Jiaying Liang[4], Jiawei Wang[1], Shanshan Song[1], Jian Gao[1], Weifeng Yang[4,5], Hongxing Xu [1] & Jian Wu [1,6,7] ✉

Nanoplasmas induced by intense laser fields have attracted enormous attention due to their accompanied spectacular physical phenomena which are vigorously expected by the community of science and industry. For instance, the energetic electrons and ions produced in laser-driven nanoplasmas are significant for the development of compact beam sources. Nevertheless, effective confinement on the propagating charged particles, which was realized through magnetic field modulation and target structure design in big facilities, are largely absent in the microscopic regime. Here we introduce a reliable scheme to provide control on the emission direction of protons generated from surface ionization in gold nanoparticles driven by intense femtosecond laser fields. The ionization level of the nanosystem provides us a knob to manipulate the characteristics of the collective proton emission. The most probable emission direction can be precisely steered by tuning the excitation strength of the laser pulses. This work opens new avenue for controlling the ion emission in nanoplasmas and can vigorously promote the fields such as development of on-chip beam sources at micro-/nano-scales.

When a nanometer-sized structure is illuminated by an intense irradiation, electrons can be stripped away from individual atoms and being partially trapped by the highly charged ionic clusters, forming transient nanoplasmas[1,2]. Compared to the conventional macroscopic plasmas, nanoplasmas exhibit unique advantages including miniaturization, integratability, and easily achievable excitation conditions. The superb characteristics of nanoplasmas have already demonstrated vast application potentials in the emerging fields of generating extreme ultraviolet light sources[3], compact particle accelerators[4], ultrafast optoelectronics[5], etc. Up to date, nanoplasmas can be produced by shining the intense optical fields onto neutral atomic aggregates of rare gas species[6,7], helium nanodroplets[8–12], and nanoparticles composed of dielectric or metal materials[13–15]. When nanostructures are coupled to optical fields,

enhanced near fields can be induced on the nanometer scales, where the enhancement can sometimes go beyond orders of magnitude[4,16]. When the driving field is strong enough, the nanosystem can be sufficiently ionized thus producing large amounts of electrons and charged ions. Meanwhile, the extreme couplings between the nanostructure and the rapidly altered external driving fields can lead to dramatic manipulation on the intrinsic properties of the nanosystem. The underlying processes such as thermal electron emission[17], inelastic scattering of Auger electrons[18], charge recombination[19,20], and avalanche ionization[8] have been revealed in nanoplasmas based on state-of-the-art techniques. People have gained a fundamental insight into the ultrafast dynamics in nanoplasma formation and relaxations. However, effective control on the emission behaviors of nanoplasma is still lacking.

[1]State Key Laboratory of Precision Spectroscopy, East China Normal University, Shanghai 200241, China. [2]School of Information Science and Engineering, Harbin Institute of Technology, Weihai 264209, China. [3]College of Science, Harbin Institute of Technology, Weihai 264209, China. [4]School of Physics and Optoelectronic Engineering, Hainan University, Haikou 570228, China. [5]Center for Theoretical Physics, Hainan University, Haikou 570228, China. [6]Chongqing Key Laboratory of Precision Optics, Chongqing Institute of East China Normal University, Chongqing 401121, China. [7]Collaborative Innovation Center of Extreme Optics, Shanxi University, Taiyuan, Shanxi 030006, China. ✉e-mail: hli@lps.ecnu.edu.cn; scjiang@lps.ecnu.edu.cn; jwu@phy.ecnu.edu.cn

In the present work, we have achieved precise control on the directional emission of protons generated in the interaction between gold nanospheres and femtosecond laser fields. As illustrated in Fig. 1, femtosecond laser pulses were interacted with a beam of flowing gold nanospheres in a vacuum chamber. The gold nanoparticles are injected into the vacuum by an aerosol source to guarantee that the particles involved in the light-and-matter interactions are kept refreshing. Sodium citrate or alcohol molecules could be attached on the surface of the gold nanospheres as they enter the interaction zone in the vacuum. Since the gold nanoparticles were sent into the vacuum chamber through a drying tube, where most of the surface molecules can be eliminated, only a small contamination can be left on the surfaces of gold nanospheres in the interaction region. The $H^+$ ions generated from the ionization of surface molecules attached to the nanospheres were detected in a velocity map imaging (VMI) spectrometer equipped with a complementary metal oxide semiconductor camera, where the momentum distributions were recorded in a single-shot manner. Because protons have the smallest mass of all ion species, the collective ejection and propagation of protons precede other ion products, making them promising for further manipulations. We find that the direction of the collective emission of protons can be continuously regulated by precisely controlling the degree of ionization of the nanosystem, the essence of which are the near-field-induced ionization and the interaction among charged particles. Particularly, when being sufficiently ionized and reaching the plasma critical density at the leading edge of the excitation laser pulse, the localized resonant absorption can cause a concentrated ion emission in the backward direction. In our work, the collective proton emission angles can be continuously tuned from "forward" to "backward" direction by optimizing the laser field parameters. This work opens new avenues for the generation and modulation of compact ion beam sources and may bring inspiration to the field of precision machining. The well-tailored ion emission can promote the applications like radiation therapy and high-precision imaging.

## Results

### Distinct ionization regimes for gold nanospheres

We explored two distinct regimes where different processes dominate in the strong-field-and-nanoparticle interactions. One is the near-field-driven ionization (NFDI) regime, where a relatively low laser intensity is applied to the gold nanoparticles. The enhanced near-field distribution which can be estimated based on finite-difference time-domain (FDTD) calculations initiates the ionization of the molecules attached to the surface of the nanoparticles. The interior fields are essentially screened in this case for a metal nanoparticle. The situation can be dramatically altered when we crank up the laser intensity. The free electron density created by the leading edge of the laser pulse around the nanoparticle can reach a critical value and form localized resonant absorption. There exist harsh requirements for the realization of a plasma beyond the critical density, involving the build-up of the dense electron ensemble that can compensate for the expansion of the system in ultrashort time scales, and specific collective oscillation phases with respect to the electric field oscillation of the driving laser[21], etc. Nevertheless, the condition can be satisfied somehow by precisely tuning the external laser field and optimizing the extreme conditions in the nanoscales. In this regime, resonant absorption can be achieved on the backside of the nanoparticle thus producing considerable surface ionizations, and photoion ejections toward the backward direction.

Based on Mie theory[22], the characteristic response of a nanoparticle in a laser field is dominated by the size parameter $\rho$, defined as $\rho = \pi d/\lambda$, where $d$ is the diameter of the nanoparticle and $\lambda$ is the excitation wavelength. A forward focus can be expected for the single-particle excitation when the size parameter is approaching or larger than unity. This applies for the NFDI regime, where the spatial profile of the near field can be estimated using the FDTD simulations. To estimate the far-field momentum distributions of the ion emission we performed the calculations based on a simple classical model[13] (details of the calculations please see the SI). The experimental and calculation results of proton emission are presented in Fig. 2 for the 400 nm excitation. We can see that the difference in the near-field profile can be manifested in the far-field momentum distributions of protons. As confirmed by previous work[23] the radial emission dominates for the protons born near the hot spots because of the high symmetry of the initial local charge distributions. In this case, a one-to-one mapping between the birth and the final emission angles can be obtained. However, for a highly asymmetric local charge distribution generated at 400 nm excitation, the actual final emission direction of protons will be deflected to a larger angle. A slight deviation in the degree of asymmetry in the near-field distribution can cause a dramatic difference in the final momentum distributions. The 400 nm excitation can cause a clear forward focusing of the $H^+$ ions at low excitation intensity. As shown in Fig. 2a, f, the momentum distributions of protons emitted from nanoparticles exhibit a clear forward focusing pattern for the 400 nm excitation at laser intensities below 10 TW cm$^{-2}$.

When we increase the excitation intensity, the response of nanosystems can be dramatically modified. Take the 400 nm excitation for example, a broad momentum distribution of the proton emission in the "backward" direction along the laser propagation begins to manifest in the momentum spectra when the laser intensity exceeds 75 TW cm$^{-2}$ (Fig. 2b, c). In our single-shot VMI spectroscopy, the size of nanoparticle is much smaller than the size of the laser focus (which is in the range of tens of microns). Due to the fact that the laser beam exhibits a Gaussian profile in space, the specific intensity experienced by each nanoparticle can be dramatically different. In general, the results at lower excitation intensities overlap with part of the results obtained at higher intensities. The momentum distributions shown in Fig. 2 are the integrated results on thousands of single-shot measurements. In principle, these images contain the results under a series of laser intensities[24]. The signal profile located in the $-y$ direction, corresponding to the "forward" direction, is generated under relatively weak laser excitation. The forward focused near-field distribution can be responsible for the forward focusing of the $H^+$ ions in their far-field momentum distributions. However, the "backward" emissions occupy the $+y$ direction in the momentum map, only appearing when the intensity is sufficiently high. We can see that the backward ion emission exhibits a maximum yield around 50 a.u. in momentum, corresponding to a proton energy of about 18.7 eV. As the intensity increases, the protons with the highest energy can even fly out of the edge of our detector. Nevertheless, we estimate a cutoff energy of around 130 eV at 100 TW cm$^{-2}$ excitation strength of the laser pulse.

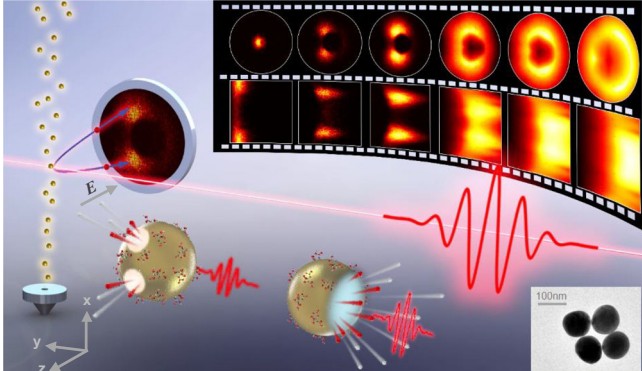

**Fig. 1 | Schematic drawing of the experimental apparatus.** Linearly polarized femtosecond laser pulses interact with isolated gold nanoparticles in a VMI chamber. By tuning the excitation laser intensity, the emission angle of proton can be tailored. The inserted image on the lower right corner is the transmission electron microscope (TEM) image for the 100 nm gold nanoparticles.

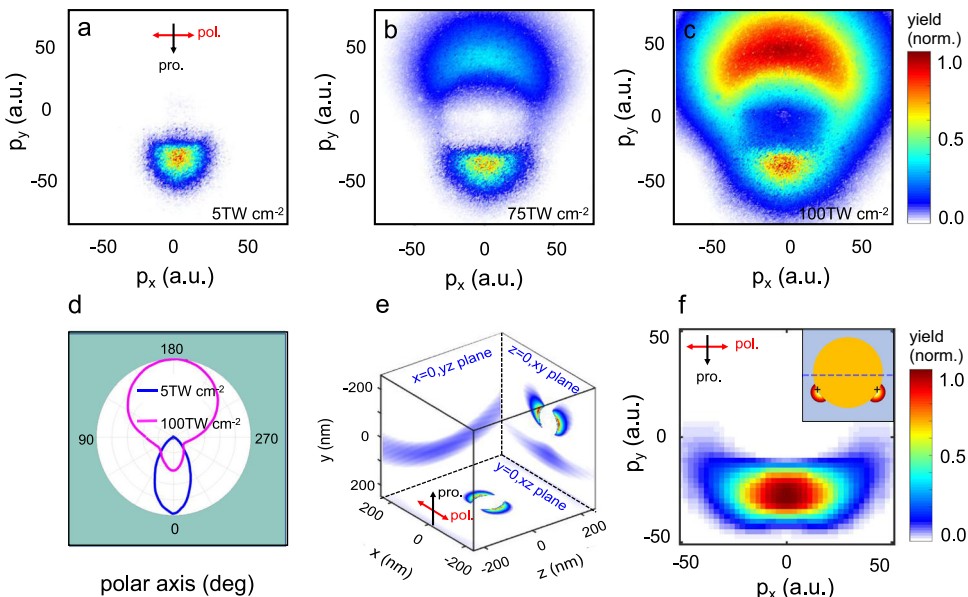

**Fig. 2 | Measured and simulated distributions of protons under different laser intensities. a–c** The integrated momentum distributions of protons generated from surface molecular ionization in gold nanosystems at the intensities of 5, 75, and 100 TW cm⁻² at 400 nm, respectively. The black and red arrows in (**a**) indicate the laser propagation and polarization directions, respectively. **d** Polar plots of the proton momentum distributions at 5 and 100 TW cm⁻². **e** Calculation results of the near-field |E| distributions based on FDTD for 100 nm gold nanospheres interacting with low-intensity laser pulses at 400 nm. **f** The calculated far-field momentum distributions of protons ejected from gold nanoparticles irradiated by 400 nm. Inserted are the initial spatial distributions of the protons from gold nanospheres. The plus signs at (−40 nm, −15 nm) and (40 nm, −15 nm) represent the effective charges for the localized plasmas.

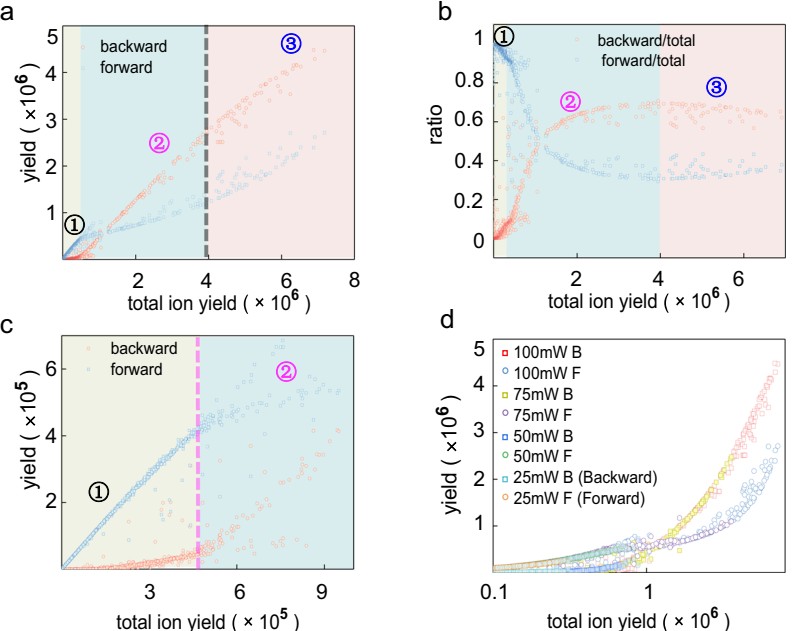

**Fig. 3 | Statistics of the forward and backward ion yields/ratios as a function of the total ion yield. a, c** Statistics of the forward and backward ion yields as a function of the total ion yield for each single-shot measurements (**c**) is a zoomed in for the region at low total ion yields. Three different regions are discussed: ① forward emission is dominated; ② forward emission growth rate is suppressed and backward emission quickly surpassing forward emission; ③ forward and backward proton yield tend to be consistent. **b** Statistics of the ratios of the forward and backward ion yields over total yields as a function of the total ion yield. **d** Statistics of forward and backward emitted protons for various excitation laser powers (the horizontal axis is displayed in log scale).

To clarify the origin of the backward emitted protons, the ion yield in the forward and the backward directions is integrated separately for each single-shot measurement and is plotted as a function of the single-shot total ion yield, where a larger single-shot total ion yield corresponds to a higher effective laser intensity. As shown in Fig. 3a, the forward and backward emitted protons follow a clear profile which

can be divided into a couple of regions. Region 1 is dominated by the forward emissions (see Fig. 3a and the zoomed plot in Fig. 3c). In this region, the small total ion yields indicate a weak ionization level in the whole nanosystem, which can be attributed to the NFDI. When the total ion yield increased up to about $5 \times 10^5$ (indicated by the magenta dashed line in Fig. 3c), the growth of the forward ion yield starts to be

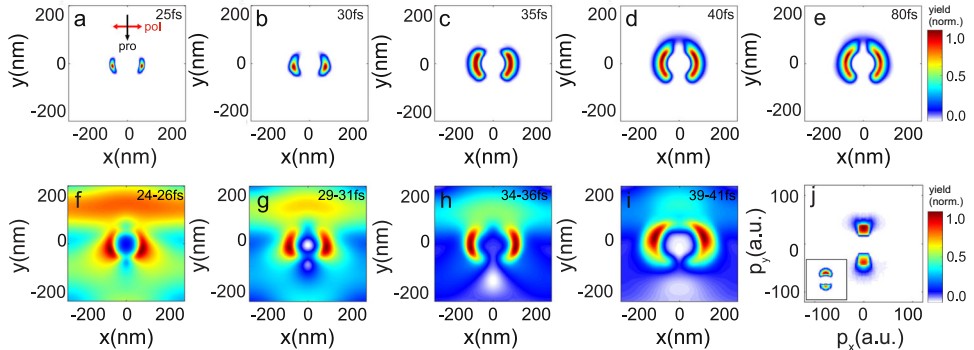

**Fig. 4 | PIC calculation results (EPOCH2D version 4.18.0). a–e** The instantaneous distributions of protons generated from surface $H_2O$ molecular ionizations in gold nanoparticles ($d = 100$ nm) at 25, 30, 35, 40 and 80 fs in the real space, respectively. The excitation laser parameters are center wavelength 400 nm, peak intensity 100 TW cm$^{-2}$, and pulse duration 30 fs (full width at half maximum). The black and the red arrows in (**a**) indicate the laser propagation and polarization directions, respectively. **f–i** Averaged distributions of the enhanced electric field around the nanoparticle for the period of 24–26 fs, 29–31 fs, 34–36 fs and 39–41 fs. **j** Calculated far-field momentum distributions for protons. The data were interpolated and smoothed due to the large difference in scale between the initial and final states. Inserted is the corresponding experimental single-shot data.

slowed down while the increment of the backward emission accelerates. This is the turning point where we can start to obtain significant backward emission signals in the momentum distributions. The growth of the backward emissions will reach a saturation when the total ion yield reaches about $4 \times 10^6$ (indicated by the gray dashed line in Fig. 3a). It has been demonstrated that the total ion yield for each single-shot measurement can be a characteristic parameter indicating the excitation level of the nanosystem, or the effective excitation field strength[25]. This is verified by our multiple data acquisitions at different excitation powers (Fig. 3d). The average power is correlated to the maximum single-shot ion yield and the lower excitation regions are largely overlapped for different sets of data. When the external laser field is increased, the near-field distribution from simple FDTD calculations will become invalid, due to the fact that the intrinsic property of the nanosystem could be altered in an ultrafast time scale. Previous studies have shown that ionization-induced metallization of dielectric nanoparticles can be initiated in intense laser fields[19,21]. The large amount of rapidly released free electrons in the nanosystem can ignite avalanching in a small volume, dramatically modifying the nanoscale environment, and a pronounced resonant near-field enhancement can occur. The situation for metal nanoparticles can be distinct such that the interior region of the particle can be screened thus the ionization may not be ignited inside the particle. How metal nanosystems change at the ultrashort time scale is still a mystery in extreme conditions.

In order to explore the dynamics when the target is irradiated by stronger laser fields, the modified particle-in-cell (PIC) numerical method based on EPOCH code package[26] is utilized. The coupled Newtonian equations for each pseudoparticle involving 100 microscopic particles and the Maxwell equation are solved in a $\lambda \times \lambda$ 2-dimensional (2D) box with CPML boundary condition. Initially, the density of the Au$^+$ ion and electrons is set to be $6 \times 10^{28}$ m$^{-3}$ inside the nanosphere, which is the density of gold at room temperature. At the surface of the nanosphere, the densities of neutral H and O atoms are set as a function $\rho(r) = 2\rho_0(1 + e^{(r-r_0)/0.01\mu m})^{-1}$ which decreases rapidly from the nanosphere surface $r_O$ to the outward. The densities of H and O are set at the ratio of 2:1. The calculated time-resolved distributions of the H$^+$ ions excited by a femtosecond laser pulse (at 400 nm central wavelength, 30 fs pulse duration, 100 TW cm$^{-2}$ peak intensity) are presented in Fig. 4a–e. The most probable location of the H$^+$ ions is in the forward direction of the laser propagation at early times (Fig. 4a), consistent with the instantaneous forward focusing near-field distribution shown in Fig. 4f. At a latter time of the leading edge of the laser, the near field and the resulting ion distributions exhibit a switching from the forward side ($-y$ direction) to the backward

direction ($+y$ direction) (Fig. 4c, h). After the laser reaches its peak, the probability of H$^+$ ions in the rear side becomes dominant (Fig. 4d, i). As the driving laser is sufficient to generate prominent nanoplasma by surface ionizations, the near-field enhancement can be changed dramatically in nanometer and femtosecond spatial-temporal scales (as shown in Fig. 4f–i). The ionization-induced modification in the field distribution can also be explained by a core-shell model in which the metal nanosphere is coated by layer of plasma, generated from the electrons ionized at the leading edge of the driving laser pulse. The density profiles of the nanoplasma change in femtosecond time scales, resulting in modification on the resonance absorption position. The far-field momentum distribution of the H$^+$ ions can be obtained from a proceeding calculated based on the classical model (details can be found in Section I of the SI), and the corresponding results are shown in Fig. 4j. The far-field momentum distribution is a propagation result of the initially localized protons under the action of Coulomb forces from the other heavier ions. The degree of asymmetry of the initial charge distributions can result in dramatically different final momentum distributions. Radial emission is more probable for protons born with higher symmetry of the local charge distributions, leading to a one-to-one mapping between the birth and final emission angles[23]. In this condition, the far-field momentum distribution can inherit the main features of the near-field distribution. However, with an asymmetric initial charge distribution, the protons would be preferred to be reflected toward larger angles, resulting in the forward and backward distribution observed at 400 nm. When the driving laser wavelength is 800 nm, the near-field enhancement is more delocalized and more symmetric with respect to the laser polarization direction compared to the cases at 400 nm. This difference leads to different far-field momentum distributions for 400 and 800 nm excitations.

More detailed theoretical results for different laser intensities are presented in Supplementary Fig. 9 in the SI and are consistent with the experimental observations shown in Fig. 2. The dominated backward ion emission agrees qualitatively with our experimental observations. In this strong-field regime, the free electron density created at the leading edge of the laser pulse can quickly reach a critical value at the backside of the nanoparticle, where dramatic resonance absorption occurs. Considerable ionizations can be induced locally and result in photoion ejections toward the backward directions. Briefly, the fundamental mechanism of process can be described by the following steps: (1) Forward emission: the external field strength is relatively low at the leading edge of laser pulse (lasting for a few optical cycles), the near-field enhancement of gold nanoparticles obeys the Mie theory, and protons are mainly emitted in the forward direction. (2) Electron

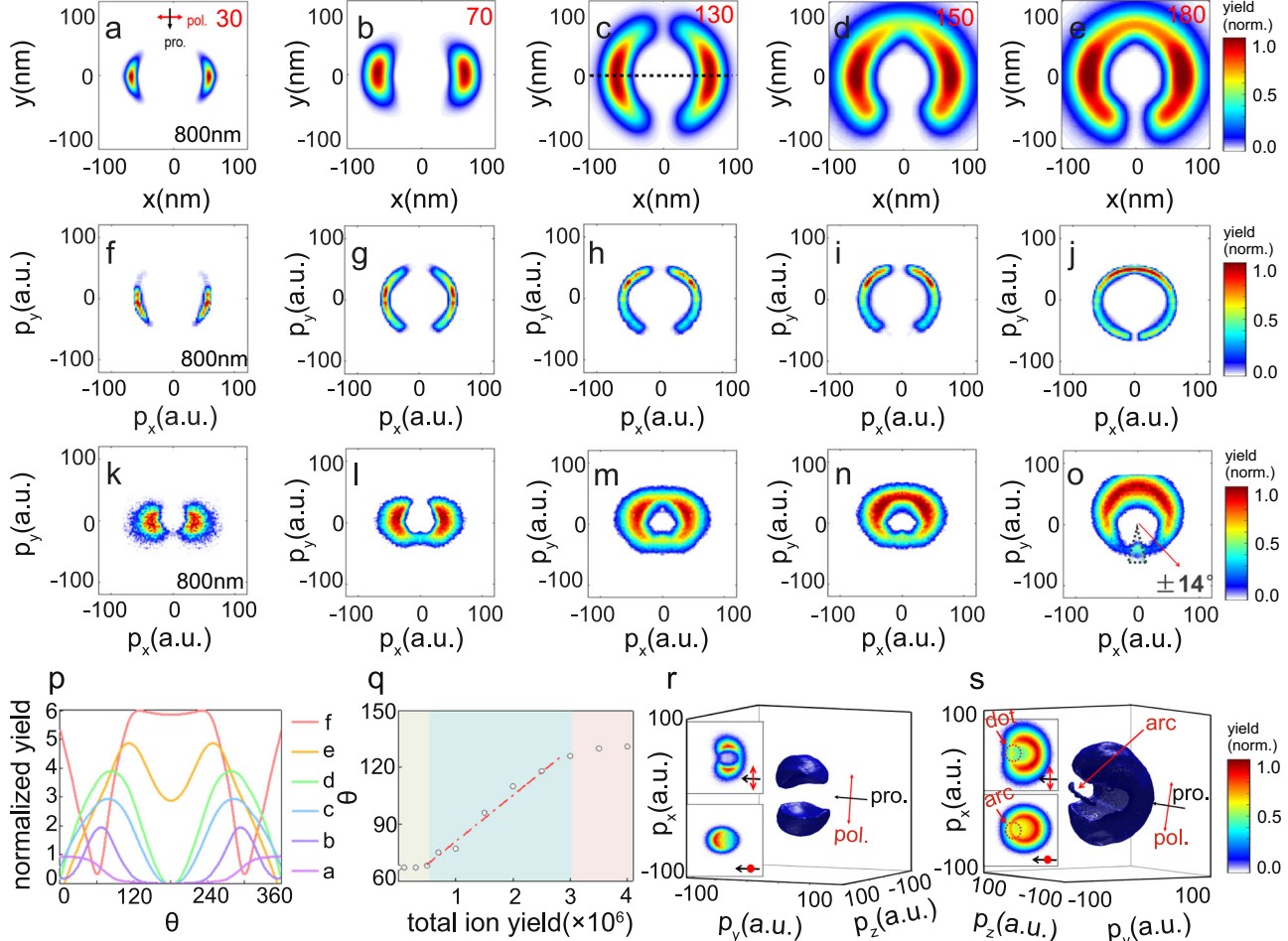

**Fig. 5 | Theoretical and experimental proton distributions at 800 nm under different laser intensities. a–e** Calculated intensity-dependent near-field real-space proton distributions at 800 nm. The unit for the value indicated in the upper right corner of each image is TW cm$^{-2}$. **f–j** and **k–o** are the calculated and measured far-field momentum distributions, respectively. A narrow proton emission within an angle range of ±14 degree is marked in (**o**). **p** The normalized angular accumulation: the laser field keeps growing with time, rapid aggregation of free electrons will occur within femtosecond time scales at the backside of the nanoparticle. (3) Resonant absorption induced ionization: the density of the electrons can reach a critical value and resonant absorption takes place at the backside of the nanosystem[15,21]. This mechanism opens a new way for the manipulation of ion emission in nanosystems. distributions of the ion yields for various excitation strengths. The data at various intensities were multiplied by a factor for better illustration. **q** The peak emission angles as a function of the total ion yields. The red dashed line is obtained by a linear fitting based on the formula indicated in the figure. **r, s** Three-dimensional isosurface of the topological measurement results obtained at two typical laser intensities, -30 and -180 TW cm$^{-2}$, respectively.

**Continuous tuning of proton emission from nanoplasmas**
To realize precise control on the directional proton emissions, femtosecond laser pulses at excitation wavelength of 800 nm are utilized. The size parameter for 800 nm is relatively small. In this condition, the near-field distribution under low intensities shows a higher symmetry along laser propagation direction compared to the cases at 400 nm. According to our calculations, a two-lobe momentum distribution along the polarization direction can be obtained at low intensities at 800 nm, with slightly forward focusing effect. Calculation results for the near-field real-space and the far-field momentum space distributions under different laser intensities can be seen in Fig. 5a–j. The most probable exit angle of protons can gradually converge to the back direction when we increase the laser intensity. This is manifested in the intensity-dependent experimental data shown in Fig. 5k–o (the data were symmetrized for the ±x direction to get rid of the

inhomogeneous detection efficiency of the detector). The different responses between 400 and 800 nm excitations prove that the initial symmetry of local charges has a significant impact on the final momentum distributions. The angular distributions of ion yields are plotted in Fig. 5p for various excitation strengths at 800 nm. The angle with the maximum yield can be continuously changed from 67° to 131° by increasing the excitation intensities. As shown in Fig. 5q, control on the ion emission direction shows a linear regime where an empirical relation can be derived: $\theta = k \times Y_{ion} + C$, where $\theta$ is the most probable emission angle for protons, and $Y_{ion}$ is the total ion yield. The fitting parameters turn out to be $k = 2.4 \times 10^{-5}$, $C = 56$. Saturation occurs for stronger excitations with a total ion yield beyond $3 \times 10^6$. This saturation effect may originate in the ionization of the nanosystem and in the detection efficiency of our experimental setup. Also, the continuous tuning is invalid when the laser intensity is too low where NFDI becomes dominated.

Interestingly, a narrow proton emission within an angle range of ±14° can be recognized in the forward direction for the result shown in Fig. 5o. To clarify this unexpected distribution, we performed a tomographic measurement on the proton emission showing the three-dimensional (3D) momentum distributions (Methods for the tomographic measurement can be found in the SI). The 3D profiles obtained for a lower and a higher laser intensity are presented in Fig. 5r, s,

respectively. In Fig. 5r, we can see the characteristic two-lobe profile, in accordance with the 2D distributions shown in Fig. 5k. However, the 3D distribution shown in Fig. 5s obtained at high intensity exhibits a narrow band. This band presents an arc or dotted distribution when projecting onto two orthogonal directions. This agrees with our observations in Fig. 5o. This narrow band proton emission generates from the Coulomb interaction from the rest of the ejected ions. Based on the 3D distribution, a large amount of protons composed a wide distribution which is elongated in the laser polarization direction. This profile can be equivalent to two effective positive charges located on the two sides along polarization. The initially ejected ions in the $-y$ direction will be repelled by these two effective charges and be focused to a narrow band after a certain time. The formation time scale should be in the regime of picoseconds. The formation dynamics of such a special profile could be of great interest for further explorations.

Using the single-shot VMI technique, the ultrafast dynamics of gold nanoparticles irradiated by strong laser field are investigated by measuring the momentum distribution of protons ionized from the surface molecular of nanoparticles. When the excitation laser intensity is within a specific range, the emission direction of protons generated from gold nanoparticle surface will change dramatically. Theoretical simulation and experimental results prove that with the increase of laser intensity, the electron yield can achieve the plasma critical density within femtosecond time scales, and strong resonance absorption starts to occur at the backside of nanoplasmas, leading to the observation of backward proton emissions at high excitation intensities. Our work demonstrates that continuous and accurate control on the directional ion emission is possible in nanosystems driven by intense femtosecond laser pulses. This effect and the underlying mechanism are essentially important for fundamental science, and show vigorous potential in the development of compact beam sources, nanoswitchs, and to promote the techniques such as laser-driven precision machining.

## Methods

The measurement was performed in an ultrahigh-vacuum chamber of VMI apparatus[27] where an aerosol source is implemented[28]. As illustrated in Fig. 1, the femtosecond laser pulses from a Ti:sapphire amplification system (35 fs @ 800 nm, 1 kHz) were focused on the flowing aerosol of 100 nm gold nanospheres by a focusing mirror ($f$ = 500 mm). The generated ions in the interactions of femtosecond laser pulses with the nanoparticle systems are guided toward a microchannel plate (MCP)/phosphor detector at the end of the spectrometer and the induced illuminations are detected by a high-speed single-shot camera (exposure time 2 μs, 1 kHz). In this work, a fast high-voltage gating with tunable time delay with respect to the laser pulse is applied to the MCP thus different ion species generated in the nanosystem can be distinguished by their arriving times on the detector. The momentum distributions shown in this work are the 2D projections of the 3D momentum distributions. The gold nanospheres used in this work are produced by seed-mediated growth technique[29] and initially dispersed in an aqueous solution. Surface attached sodium citrate is used during the preparation process to reduce the probability of cluster deposition, which composes a major part of the surface-attached molecules. Besides, solution molecules could also be attached to the surface when they enter the interaction region. The concentration of gold nanoparticle is diluted to 1 mg/L in alcohol. The nanoparticles are aerosolized by the atomizer with Ar gas (1.5 bar) and injected into the interaction chamber by an aerodynamic lens system. The angle-resolved momentum distributions of the proton from ionization of the sodium citrate or alcohol molecules attaching on the surface of the nanospheres are measured in a single-shot manner, where individual nanoparticles interact with an isolated ultrashort laser pulse. The intensities of the linearly polarized laser fields in the reaction region were estimated by examining the momentum shifts of the above threshold ionization photoelectrons from strong-field ionization of xenon atoms along the laser polarization direction[30,31]. Atomic units (a.u.) are used throughout this paper unless otherwise indicated.

## Data availability

The source data that support the main figures within this article are available from the Zenodo database[32].

## Code availability

All the codes that support the findings of this study are available from the corresponding author upon request.

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

## Acknowledgements

We thank Dr. Jianhui Bin for fruitful discussions. This work was supported by the National Natural Science Fund (Grants No. 92050105, 12227807, 92250301, 12241407, U22A2008, 12204132, 12304376); Excellent Youth Science Foundation of Shandong Province (Overseas) (2022HWYQ-073); Fundamental Research Funds for the Central Universities (HIT.OCEF.2022042). The work was also supported by the Shanghai Committee of Science and Technology, China (Grants No. 22ZR1419700, 23JC1402000); Shanghai Pilot Program for Basic Research; Shanghai Municipal Science; and Technology Major Project.

## Author contributions

H.L. and J. Wu conceived the idea and initiated the study. F.S., Q.Q., and J. Wu designed and carried out the experiments. S.J., F.S., Q.Q., Q.L., J.G., S.B., Y.P., J.L., J. Wang, S.S., H.X., and W.Y. performed the simulations. F.S., Q.Q., and H.L. contributed to the data analysis and writing the manuscript. F.S. and Q.Q. contributed equally to this work.

## Competing interests

The authors declare no competing interests.
