## [Peer Review File · Nature Communications]

All-optical steering on the proton emission in laser-induced nanoplasmasReviewers' comments:

Reviewer #1 (Remarks to the Author):

Fenghao Sun and coworkers present velocity-map imaging results of gold nanoparticles irradiated with high-intensity femtosecond laser pulses in an intensity regime where a high ionization rate and accordingly nanoplasma formation is expected. The authors focus on the anisotropy of the emission of protons which should stem from residual solvent molecules from the growth and aerosolizing processes. Pronounced backward-forward directions depending on the laser pulse intensities are observed in the case of 400nm wavelength; in the case of 800nm laser pulses a continuous change of preferred emission direction is observed. Backward intensities are mainly interpreted in terms of the formation of a reflecting plasma layer at high intensities. Since the individual "seen" pulse intensity always spans the laser intensity profile, sorting with respect to total detected charges has been done in order to somehow deconvolve the intensity uncertainty.

Although anisotropy effects in the interaction of femtosecond laser pulses with nanoparticles have been observed in different materials, in general, the here detected anisotropies constitute a very interesting observation. However, the manuscript is not eligible for publication because of serious deficiencies in several directions: The structure of the manuscript, the given details of the performed experiment, the presentation of data/figures, a sound interpretation, language.

In the following exemplified remarks, not covering everything one could comment on:

The structure of the manuscript does not present the material in logical way. First (page 4) there is an explanation/interpretation of different situations in nanoplasmas created in different laser field strength, without having any information what has been done in the experiment. It would be better first to introduce what has been measured and then try to do an interpretation of the findings.

There are details missing on experimental aspects. In velocity-map imaging one normally measures 2D-projections, unless one does slicing techniques by fast switching of voltages. Apparently, switching is done in the current experiment in order to select different masses. It is unclear what momentum distributions are plotted in the end (slices, projections, cuts). Moreover, only distributions of protons are shown. Is this a major contribution of ions, does one detect other masses, are these well separated in the time-of-flight, do these also show anisotropies? How are energies calibrated?

Because one only observes surfactant fragments from the surface of the metal particle it would be important to know more the composition. Does one have only a small contamination of gold surface or are the metal particles covered by thick layers? Is there information contained in the mass spectra? What does "nanoparticle dimers" mean, how are they bound, is there a surfactant layer separating the particles?

The interpretation does not present a clear picture. The significant differences between the use of different wavelength is not clear. The FDTD calculations just model field distributions. The distribution of ejected ions is influenced also by other parameters like e.g. the distribution of different materials, involved energies, ionization rates, recombination, etc.. Is there a connection to anisotropies one has measured in other systems (also dielectric nanoparticles, aerosols)? It would be desirable to model the ejection of protons in the framework of molecular dynamics simulation in order to understand the underlying mechanisms of the directionality.

In terms of the presentation of results and figures: e.g., in Fig. 5 there should be correct units and corresponding labels. Labels in the figures give ionization numbers, which I would expect to be absolute numbers which may not make much sense. On the other hand, in the text sometimes the same quantities are also called ionization rates. Furthermore, different parameters for different curves/panels in the figures must be indicated in the plot or in the caption. In the 3D-distributions in Fig. 5 there is no 3D information to be identified.

In terms of language there are numerous mistakes: "There is possibility that the free", "the protons with the highest energy at cutoff increases and can even flying out", "How metal nanosystems changes at the ultrashot time", ...

In conclusion, the manuscript is not in a status to be considered for publication.

Reviewer #2 (Remarks to the Author):

This manuscript investigates angular distribution of proton emissions from gold nanoparticles under femtosecond pulse laser radiation. The authors show that one can continuously change the emission direction from "forward" to "backward" of the nanoparticle by optimizing the radiation laser field. Utilizing theoretical simulations and a single-shot velocity map imaging (VMI) technique, the authors explained this change as the effect of a "plasma mirror" formed when the electron yield of nanoparticles exceeds the plasma critical density due to strong excitation.

It seems to me that this manuscript is a successor paper to previous study by the authors (Ref. [13,14]). The basic setup and analysis methods are similar. The advantage of this manuscript is that it confirms the dominance of "backward" proton emission due to the "plasma mirror" effect and proposes its cause. However, compared to FDTD simulations in the weakly excited regime, the estimate of the electron yield in the strongly excited regime is very rough. It does not seem to me that this estimate sufficiently explains the femtosecond dynamics and local resonant absorption of electrons in metallic nanoparticles under strong excitation. So, I do not recommend this manuscript for publication in Nature Communications.

Minor comments, not a major factor in the recommendation, are attached below.

(1)

Figure 1 shows that the laser beam propagates along the x-direction, but it appears to propagate in the y-direction in Figure 2 and later.

(2)

The manuscript states that the proton is due to ionization of sodium citrate and ethanol on the metal nanoparticle surface, but does it not emit other ions such as sodium ions? If so, does their directional dependence of the emission appear in the same way as that of the protons?

(3)

The vertical and horizontal axes in Figure 2c are scaled at 10^6 , but in this figure they are actually scaled at 10^5 .

(4)

In Figure 2b, as the total ion yield increases, the forward and backward components appear to approach each other again. Can this be explained by the "plasma mirror" effect?

Reviewer #3 (Remarks to the Author):

In this paper, the authors measured proton emission profile from gold nanoparticles pumped by intense optical excitation. The major discovery is an emission direction dependence on the pump power. With the increase of excitation power, an intense backward emission is observed. With a semiclassical model, the author argues that the backward emission results from the plasmonic mirror effect of the ionized gold atoms. The data are clearly presented, and arguments can be easily followed. The discovery could potentially provide an efficient method to control proton emission. I have a few technical questions I hope the authors to address.

1. The axis labeling in the schematics of Fig. 1 and Fig. 2 is confusing. Can the authors more explicitly explain the arrows indicated in Fig. 2 and subsequent figures? If the polarization of the laser is indicated by the red arrow along x axis, is it in line with Fig. 1?

2. The near-field and emission simulation at low pump power shows that 800nm should emit along both positive and negative p_x direction (Fig. 2). However, the supplementary figure S3 and Fig. 5

shows that, the 800nm emission is also slightly but obviously forward ($-p_y$) leaning. Looking at the 400nm results in Fig. 3 and Fig. S3, the forward focusing angle is much smaller (closer to the $-p_y$ direction) compared to the simulation in Fig. 2. How does the simulation reconcile with this observation at low intensity?

3. The data in Fig. 5a to 5g seems almost perfectly symmetric with respect to p_y axis. Can the authors provide higher quality figures and indicate whether there is symmetrization to the data?

4. The additional small forward emission observed in Fig. 5g seems not reflected at small angles (around 0 or 360 degree) in Fig. 5h curve I7. Can the authors comment on the consistency between Fig. 5h and 5g?

5. Please indicate the unit of the physical quantity in Fig. 2. Is it absolute value of near-field electric field $|E|$ or power E^2 in Fig. 2a and 2b?

Response to Referee 1

Comments to the Author

Fenghao Sun and coworkers present velocity-map imaging results of gold nanoparticles irradiated with high-intensity femtosecond laser pulses in an intensity regime where a high ionization rate and accordingly nanoplasma formation is expected. The authors focus on the anisotropy of the emission of protons which should stem from residual solvent molecules from the growth and aerosolizing processes. Pronounced backward-forward directions depending on the laser pulse intensities are observed in the case of 400nm wavelength; in the case of 800nm laser pulses a continuous change of preferred emission direction is observed. Backward intensities are mainly interpreted in terms of the formation of a reflecting plasma layer at high intensities. Since the individual “seen” pulse intensity always spans the laser intensity profile, sorting with respect to total detected charges has been done in order to somehow deconvolve the intensity uncertainty.

Although anisotropy effects in the interaction of femtosecond laser pulses with nanoparticles have been observed in different materials, in general, the here detected anisotropies constitute a very interesting observation. However, the manuscript is not eligible for publication because of serious deficiencies in several directions: The structure of the manuscript, the given details of the performed experiment, the presentation of data/figures, a sound interpretation, language.

We thank the Referee very much for his/her careful review and the kind comment of ‘interesting’ of our work. We also thank him/her for pointing out the weaknesses of our previously submitted manuscript which helped us to make improvements. According to the comments and suggestions, we have made major modifications on the manuscript, involving adding new theoretical results to support the forward-to-backward directional control of the surface ion emissions. We believe the newly submitted work possess sufficient significance and impact for publication.

1. The structure of the manuscript does not present the material in logical way. First (page 4) there is an explanation/interpretation of different situations in nanoplasmas created in different laser field strength, without having any information what has been done in the experiment. It would be better first to introduce what has been measured and then try to do an interpretation of the findings.

We thank the Referee very much for this comment. We have made proper adjustments in the revised manuscript. The following sentences have been added at the end of Page 4.

“As illustrated in Fig. 1, femtosecond laser pulses were interacted with a beam of flowing gold nanospheres in a vacuum chamber. The gold nanoparticles are injected into the vacuum by an aerosol source to guarantee that the particles involved in the light-and-matter interactions are kept refreshing. Sodium citrate or alcohol molecules could be attached on the surface of the gold nanospheres as they enter the interaction zone in the vacuum. Since the gold nanoparticles were sent into the vacuum chamber through a drying tube, where most of surface molecules can be eliminated, only a small contamination can be left on the surfaces of gold nanospheres in the interaction region. The H^+ ions generated from the ionization of surface molecules (mainly composed of sodium citrate

and ethanol) attached to the nanospheres were detected in a velocity map imaging (VMI) spectrometer equipped with a complementary metal oxide semiconductor (CMOS) camera, where the momentum distributions were recorded in a sing-shot manner.”

2. There are details missing on experimental aspects. In velocity-map imaging one normally measures 2D-projections, unless one does slicing techniques by fast switching of voltages. Apparently, switching is done in the current experiment in order to select different masses. It is unclear what momentum distributions are plotted in the end (slices, projections, cuts). Moreover, only distributions of protons are shown. Is this a major contribution of ions, does one detect other masses, are these well separated in the time-of-flight, do these also show anisotropies? How are energies calibrated?

We thank the Referee for this comment. He/She is right that switching is done in the current experiment in order to select different masses. The high voltage switch was applied on the MCP detector at a proper time delay to select the H^+ ion signals. What have been plotted in this manuscript are the 2D-projections of the 3D momentum distributions integrated over the time window of the high voltage switch. This point has been clarified in the Section of “Materials and Methods” in the revised manuscript. The sentence has been added.

“The momentum distributions shown in this work are the 2D projections of the 3D momentum distributions.”

In the present work, we have also performed 3D tomographic imaging. The detailed method has been described in details in the SI. The experiment was conducted by multiple data obtained at different laser polarization directions and succeeding reconstruction analysis based on iradon transform algorithm.

As for the contribution of various ions in the experiment, we have added a new section in the SI to illustrate this point. In the time of flight spectrum (presented as Fig. S7 in the SI and below as Fig. R1), a major contribution from H^+ ions and other ion species contributed by the CH_n^+ groups at around $m/q=12$ can be obtained. The yield ratio of the CH_n^+ groups to that of the H^+ ions can be varied from shot to shot, but mostly in the level of a few percent. Therefore we only present the results from the major contribution, i.e. the H^+ ions, in the main text. Due to the fact that the directional ion emission is caused by the local field distribution and transient absorption enhancement, the anisotropies observed for the H^+ ion can also be observed for the other ion species from surface ionization. The momentum distributions for the ions at around $m/q=12$ have been presented in Fig. S8 in the SI, showing clear backward emission behavior at high laser intensities.

Figure R1. The single-shot time of flight (TOF) spectrum obtained for the 100 nm gold nanospheres excited by intense femtosecond pulses.

Energy is calibrated based on a measurement of the ATI electrons from Xe gases. The energy gap between the adjacent ATI peaks equals the photon energy, i.e. about 1.55 eV at 800 nm excitations.

3. *Because one only observes surfactant fragments from the surface of the metal particle it would be important to know more the composition. Does one have only a small contamination of gold surface or are the metal particles covered by thick layers? Is there information contained in the mass spectra? What does “nanoparticle dimers” mean, how are they bound, is there a surfactant layer separating the particles?*

The surface of nanostructures is inevitably covered by certain surfactant fragments during the processes of, e.g. fabrication and conservation. In our experiment, the gold nanoparticles were sent into the vacuum chamber through a drying tube, where most of surface molecules can be eliminated. Based on this and our observations, we think that only “a small contamination” can be left on the surfaces of gold nanospheres in the interaction region. In addition, only a small amount of ion species other than proton can be obtained in the time of flight measurement (mass spectrum has been added in Section IV in the SI), demonstrating a small amount of contamination on the surface. This can also be confirmed in other works investigating the surface ionization products in nanosystems using similar aerosol sources [ACS Photonics 7, 1885 (2020). Nat. Commun. 12, 3839 (2021). Nanophotonics 10, 2651 (2021).].

To make it clear to the readers, we have added the following sentences on Page 5 in the main text.

“Since the gold nanoparticles were sent into the vacuum chamber through a drying tube, where most of surface molecules can be eliminated, only a small contamination can be left on the surfaces of gold nanospheres in the interaction region.”

Nanoparticle dimers represent the adhesion of two gold nanospheres, the strong-field response of which has been reported in former works [ACS Photonics 7, 1885 (2020). Nanophotonics 10, 2651 (2021).]. Although the gold nanoparticles contain surface functional groups of sodium citrate to prevent adhesion, there may still be a small amount of dimer presented in our sample. Due to the

fact that dimer systems possess stronger near field enhancement compared to single nanoparticle, they can be excited at very low laser intensity which is not sufficient to excite the ionization processes in single nanospheres. We estimate that there can be a surfactant “layer” separating the nanospheres, in the thickness of a few atoms.

4. The interpretation does not present a clear picture. The significant differences between the use of different wavelength is not clear. The FDTD calculations just model field distributions. The distribution of ejected ions is influenced also by other parameters like e.g. the distribution of different materials, involved energies, ionization rates, recombination, etc.. Is there a connection to anisotropies one has measured in other systems (also dielectric nanoparticles, aerosols)?

We thank the Referee for this question. Based on Mie theory, the characteristic response of a nanoparticle in a laser field is dominated by the size parameter ρ , defined as $\rho = \pi d/\lambda$, where d is the diameter of the nanoparticle and λ is the excitation wavelength. A forward focusing can be expected for the single-particle excitation when the size parameter is approaching or larger than unity. According to this, the field asymmetry in the propagation direction is tiny for the case at 800 nm excitation compared to that at 400 nm. To reveal the underlying mechanism of single beam manipulation on the ion emission direction, we choose 400 nm as the excitation wavelength where clear forward focusing effect exists at low intensities. When the excitation intensity was increased, the observed most probable emission direction for protons show a forward to backward switching. However, to realize continuous tuning on the directional proton emissions, femtosecond laser pulses at excitation wavelength of 800 nm are utilized. From one hand, the number of photon required for single ionization is larger for 800 nm excitation compared to that for 400 nm, providing a larger scope for tuning the ionization level of the gold nanosystems. On the other hand, the size parameter for 800 nm is relatively small. In this condition, the near field distribution under low laser intensities show a very weak forward focusing effect, which can result in a two-lobe momentum distribution, leading to proton distributions along the laser polarization direction. These factors have been clarified in the revised manuscript. Moreover, a detailed comparison of the near field enhancement and the resulting far field momentum distributions at these two distinct excitation wavelengths has been introduced in Section I in the SI.

The Referee is right that the distributions of ejected ions can be influenced by many factors other than the field distributions, e.g. the distribution of different materials, ionization rates, field-driven electron recombination, etc.. The underlying dynamics is yet to be explored. There are still large difficulties to clarify all these factors in the extreme interactions. However, a simple classical model has been developed in our early work [Nanophotonics 10, 2651 (2021)] to calculate the far-field momentum distributions under at low excitation intensities. Although a lot of approximations were involved in the calculation, the results can still provide reasonable information for related studies. The details have been provided in Section I in the SI. While for the cases at high intensity excitation, the property of the nanosystem changes dramatically in femtosecond time scales providing enormous challenges for theoretical modeling. **In the revised work we have developed a modified PIC model to reveal the underlying dynamics in such regime.**

Due to the lower ionization potential of metal materials, a large amount of electrons and ions can be produced at relatively low excitation laser intensities ($\sim 10^{14}$ Wcm⁻²) in gold nanoparticles where

the critical density can be achieved in the transient nanoplasma. In principle, the backward surface ion emission behavior can be observed in any nanoplasma systems (including dielectric nanosystems) as long as the localized critical density is reached. Our theoretical simulation speculates that the critical density can be also achieved in SiO₂ nanospheres when the excitation intensity reaches 10¹⁸ Wcm⁻². In this situation, the forward to backward tuning of the surface ion emissions can be expected. However, the required laser intensity is not achievable in our lab.

To make it clear to the readers, we have added new theoretical results based on the modified PIC model to explain the underlying dynamics in the high intensity regime. The new results have been added in Pages 9-10.

5. It would be desirable to model the ejection of protons in the framework of molecular dynamics simulation in order to understand the underlying mechanisms of the directionality.

By following the kind suggestions of the Referee, we have developed a modified particle in cell (PIC) model to calculate the underlying dynamics of light and nanoparticle interactions, including the near field distributions, the surface molecular dynamics and the succeeding propagation of the generated ions. The results can provide solid support for our observations, i.e. the directional control on the surface ion emissions by tuning excitation strength. In the revised manuscript, we have added a detailed description on the PIC calculations and the underlying mechanism on Pages 9-10. The dynamics in femtosecond time scales have been presented in Fig. 4 in the revised manuscript.

6. In terms of the presentation of results and figures: e.g., in Fig. 5 there should be correct units and corresponding labels. Labels in the figures give ionization numbers, which I would expect to be absolute numbers which may not make much sense.

We thank the Referee very much for pointing this out. The aim of providing ionization numbers is to indicate the relative excitation laser intensity. According to what has been presented in Fig. 3, the obtained ion yield can be an indication of the excitation laser intensity in each laser and nanoparticle interaction. The larger yield corresponds to a relatively higher laser intensity. We have made proper adjustments in Figure 5.

7. On the other hand, in the text sometimes the same quantities are also called ionization rates. Furthermore, different parameters for different curves/panels in the figures must be indicated in the plot or in the caption. In the 3D-distributions in Fig. 5 there is no 3D information to be identified.

We thank the Referee very much for pointing this out. We have changed the "ionization rates" to "ionization numbers" in the revised version.

To make the 3D information clear, we have replotted Figs. 5i and 5j using the isosurfaces. In the revised Fig. 5j there is a clear narrow distribution indicated by the red arrow. The projection of this structure shows a dot in the x-y plane, which are pointed out in the inserted figure. We are sorry that the former presentation cannot show this structure clearly. The structure can be seen in the newly presented figure.

8. *In terms of language there are numerous mistakes: “There is possibility that the free”, “the protons with the highest energy at cutoff increases and can even flying out”, “How metal nanosystems changes at the ultrashot time”*

We thank the Referee very much for pointing these out. We have checked the language throughout the pages and have made corrections accordingly.

Response to Referee 2

This manuscript investigates angular distribution of proton emissions from gold nanoparticles under femtosecond pulse laser radiation. The authors show that one can continuously change the emission direction from "forward" to "backward" of the nanoparticle by optimizing the radiation laser field. Utilizing theoretical simulations and a single-shot velocity map imaging (VMI) technique, the authors explained this change as the effect of a "plasma mirror" formed when the electron yield of nanoparticles exceeds the plasma critical density due to strong excitation.

It seems to me that this manuscript is a successor paper to previous study by the authors (Ref. [13,14]). The basic setup and analysis methods are similar. The advantage of this manuscript is that it confirms the dominance of "backward" proton emission due to the "plasma mirror" effect and proposes its cause. However, compared to FDTD simulations in the weakly excited regime, the estimate of the electron yield in the strongly excited regime is very rough. It does not seem to me that this estimate sufficiently explains the femtosecond dynamics and local resonant absorption of electrons in metallic nanoparticles under strong excitation. So, I do not recommend this manuscript for publication in Nature Communications.

We thank the Referee very much for his/her careful review and the helpful suggestions. We agree with the Referee that the underlying dynamics was not sufficiently explained in the previously submitted manuscript. Due to the fact that the intrinsic property of the nanosystem can be dramatically modified in very short time scales under laser excitations of high intensity, the underlying dynamics are yet to be explored such as the ultrafast changes in the near field distributions, energy transfer, and the many particle interactions, etc.. These make the theoretical modeling facing great difficulties. Nevertheless, inspired by the Referee, we have developed a modified particle in cell (PIC) model for this high intensity regime, to reveal the femtosecond dynamics and the underlying mechanism causing the directional proton emissions. In the revised manuscript, new theoretical results have been added to understand the underlying mechanism and support our observations of the forward to backward transfer of the proton emission under various laser intensities. Calculation results demonstrated that the local resonant absorption at the backside of the nanoplasma was responsible for the backward proton emissions. This can be only achieved when the local charge density reaches a critical density, corresponding to the critical density to form "plasma mirror" effect in macroscopic scales. However, reflection of the optical wave can not be observed in this sub-wavelength regime. Based on these theoretical and experimental explorations, the underlying dynamics of nanoparticle interacted with femtosecond laser fields have been revealed for a new intensity regime, where nanoplasmas can be induced and the local density distributions can be well-tailored by tuning the external laser fields. We are now confident that work provides a breakthrough for the understanding of light and nanoparticle interactions and potential control on the surface ionization processes. We hope the Referee will recommend the revised version of the manuscript with the new added theoretical results for publication in Nature Communications.

Minor comments, not a major factor in the recommendation, are attached below.

1. Figure 1 shows that the laser beam propagates along the x-direction, but it appears to propagate in the y-direction in Figure 2 and later.

We thank the Referee very much for pointing this out. He/She is right that the laser propagates along the y-direction. We have made corrections in Figure 1 in the revised version.

2. The manuscript states that the proton is due to ionization of sodium citrate and ethanol on the metal nanoparticle surface, but does it not emit other ions such as sodium ions? If so, does their directional dependence of the emission appear in the same way as that of the protons?

We thank the Referee very much for this question. This is indeed a very good question.

We have observed other ion species other than proton in the experiment. For instance, in the time of flight (TOF) spectrum (presented as Fig. S7 in the SI and below as Fig. R2), a major contribution from H^+ ions and other ion species contributed by the CH_n^+ groups at around $m/q=12$ can be obtained for a single-shot measurement. The yield ratio of the CH_n^+ groups to that of the H^+ ions can be varied from shot to shot, but mostly in the level of a few percent. Therefore we only present the results from the major contribution, i.e. the H^+ ions, in the main text. Due to the fact that the directional ion emission is caused by the local field distribution and transient absorption enhancement, the anisotropies observed for the H^+ ion can also be observed for the other ion species from surface ionization. We have added a new section in the SI to illustrate this point. The momentum distributions for the ions at around $m/q=12$ have been presented in Fig. S8 in the SI, showing clear backward emission behavior at high intensities.

Figure R2. The single-shot time of flight (TOF) spectrum obtained for the 100 nm gold nanospheres excited by intense femtosecond pulses.

3. The vertical and horizontal axes in Figure 2c are scaled at 10^6 , but in this figure they are actually scaled at 10^5 .

We thank the Referee very much for the careful reading. The labels have been corrected and has been shown in Fig. 3 in the revised manuscript.

4. In Figure 2b, as the total ion yield increases, the forward and backward components appear to approach each other again. Can this be explained by the "plasma mirror" effect?

We thank the Referee for this interesting question.

In our previous manuscript, we attributed the backward proton emission to the “plasma mirror” effect. This effect can cause a reflection of the optical wave when the plasma density reaches a critical density. However, we have performed simulations based on our modified PIC model, the propagation of the optical wave was explicitly calculated. The results show that optical waves do not reflect in this sub-wavelength scale. Thus we think that it might be not proper to use the word “plasma mirror”. However, if we increase the plasma size to a macroscopic scale under similar density in the modeling, the plasma mirror effect can be obtained. In the revised manuscript, we have changed the wording for the mechanism in high laser intensity regime. It is attributed to a local resonant absorption at the back side of the nanoplasma when a critical density can be reached. The underlying femtosecond dynamics have been revealed in the newly submitted version.

As for the phenomena pointed out by the Referee, we think that ionization saturation can be reached in the nanosystem at very high laser intensities. This might cause the forward and backward ion yield approach each other. However, in our experiment, part of the backward ions can escape from the detector under high excitation intensities, as can be seen in Fig. 2c. In this case the dots plotted in Fig. 3b exhibit deviation from the real situation. The true situation for the red circles should be in a somewhat higher ratio compared to what have been shown in Fig. 3b.

Response to Referee 3

In this paper, the authors measured proton emission profile from gold nanoparticles pumped by intense optical excitation. The major discovery is an emission direction dependence on the pump power. With the increase of excitation power, an intense backward emission is observed. With a semiclassical model, the author argues that the backward emission results from the plasmonic mirror effect of the ionized gold atoms. The data are clearly presented, and arguments can be easily followed. The discovery could potentially provide an efficient method to control proton emission. I have a few technical questions I hope the authors to address.

We thank the Referee very much for his/her careful review and the helpful comments. In the revised version, we have added new theoretical results based on a modified particle in cell (PIC) model to provide solid explanations for the femtosecond dynamics and local resonant absorption in metallic nanoparticles under strong excitations. We are now confident that work provides a breakthrough for the understanding of light and nanoparticle interactions and potential control on the surface ionization processes.

1. The axis labeling in the schematics of Fig. 1 and Fig. 2 is confusing. Can the authors more explicitly explain the arrows indicated in Fig. 2 and subsequent figures? If the polarization of the laser is indicated by the red arrow along x axis, is it in line with Fig. 1?

We thank the Referee very much for pointing this out. The labels have been modified and proper explanations have been added in the revised manuscript. The arrows have been corrected.

2. The near-field and emission simulation at low pump power shows that 800nm should emit along both positive and negative p_x direction (Fig. 2). However, the supplementary figure S3 and Fig. 5 shows that, the 800nm emission is also slightly but obviously forward ($-p_y$) leaning. Looking at the 400nm results in Fig. 3 and Fig. S3, the forward focusing angle is much smaller (closer to the $-p_y$ direction) compared to the simulation in Fig. 2. How does the simulation reconcile with this observation at low intensity?

We thank the Referee very much for this question.

According to Mie theory, the characteristic response of a nanosphere in a laser field can be dominated by the size parameter in this low intensity regime. Forward focusing effect can be expected when the size of the nanosphere is approaching the incident wavelength. Gold nanospheres with diameter of about 100 nm were used in our study. Then the forward focusing effect should be more remarkable at the 400 nm excitation compared to that at 800 nm excitation. The results based on FDTD calculation (shown in Figs. 2a and 2b) in the previous version demonstrated the above features. In the field distributions calculated for the 800 nm excitation, it seems that the near field occupies along the x-direction. However, there is still tiny deviation towards the forward direction ($-y$ direction). We have performed calculations using a classical model to calculate the far-field ion momentum distributions. In the calculations we start from the FDTD code to obtain an initial near

field distribution. Then we estimate the generated ion distributions based on the field profile. The ions flying towards the detector are calculated considering Coulomb interactions in an approximated manner. The Coulomb effect caused by ion aggregation can cause certain changes in the final momentum distributions. The far field results show a clear forward focusing (mainly occupies around the -y direction) for the 400 nm excitation and a weaker focusing behavior showing two lob distribution towards the -y direction for the 800 nm excitation. **The details of these calculations have been added in Section I in the SI.** Besides the near field distribution, the final momentum distribution also depends on the local ionization processes, the interactions among many particles when they flying towards the far field. Our simplified classical model can provide a fundamental understanding of the main properties of the light and nanoparticle interaction in this low intensity regime.

Experimental observations for various intensities were presented in Fig. S3 in the revised version. The data obtained at low intensities agree well with the calculation results shown in Fig. S1. Except for the data obtained at the lowest intensity for 800 nm (the first figure on the third row in Fig. S3), where a clear forward focusing can be recognized. This distribution should be from excitation of a dimer system consisting two nanospheres attaching to each other. The induced near field enhancement in dimer system is stronger in dimers compared to isolated nanospheres. Therefore at the lowest intensity, only dimers can generated surface molecular ionizations. The surface ionization can not be excited in isolated nanoparticles at this intensity level. This can be demonstrated by previous studies, i.e. ACS Photonics 7, 1885 (2020); Nanophotonics 10, 2651 (2021).

3. The data in Fig. 5a to 5g seems almost perfectly symmetric with respect to p_y axis. Can the authors provide higher quality figures and indicate whether there is symmetrization to the data?

We thank the Referee very much for pointing this out. The figures presented in Figs. 5(a-g) were symmetrized for along the x direction. Due to the inhomogeneous efficiency of our detector in the VMI spectrometer, the raw data show clear defects in certain regions. Two raw data are presented below. The symmetrization processing were processed to get rid of these defects. Moreover, the momentum distribution should be symmetry in the +x and -x direction in principle. We have clarified this point in the caption of Fig. 5 in the revised manuscript.

Figure. R3 Raw data of the momentum distributions.

4. The additional small forward emission observed in Fig. 5g seems not reflected at small angles

(around 0 or 360 degree) in Fig. 5h curve I7. Can the authors comment on the consistency between Fig. 5h and 5g?

We thank the Referee very much for the careful reading. The curves presented in Fig. 5h in the previous manuscript do not exhibit one-to-one correspondence to what were shown in Figs. 5a-g. The curves have been replotted in the revised manuscript. In the revised Figure 5 people can see the proton emission within a small angular range in Fig. 5f and 5h.

5. Please indicate the unit of the physical quantity in Fig. 2. Is it absolute value of near-field electric field $|E|$ or power E^2 in Fig. 2a and 2b?

We thank the Referee very much for pointing this out. The physical quantity presented in Figs. 2a and 2b is the electric field $|E|$ of the near field. In the revised version, this figure has been moved to Section I in the SI. A proper explanation has been provided in the caption of Fig. S1.

REVIEWER COMMENTS

Reviewer #1 (Remarks to the Author):

The authors have significantly improved the manuscript and taken care of the raised criticism. In particular, a modified particle in cell calculation is presented including the propagation of ions, which in principle should provide a direct comparison to the measured distributions. However, the shown results of the calculation raise questions on the interpretation and do not provide a conclusive understanding of the process. The distributions of protons (Fig. 4 a-e) show a preferred direction along the laser polarization axis (sideways with respect to the laser propagation) which even in panel d and e still has minima in the forward and backward direction. One would expect this preference along the laser polarization as also the electric field enhancement (Fig 4 f-i) clearly shows this behavior. On the other hand, the calculated far field enhancement is surprisingly strongly forward-backward peaked. I would expect that from the simulations one should get an idea what is the main mechanism to do this. Furthermore, there are no calculated results for different intensities to be compared with the measured distributions, indicating the right behavior of changing the emission direction. In contrast, it is not clear at all why the calculated distribution is so much peaked in the forward-backward direction, and how to understand the results for the different laser conditions. In the end, the interpretation in terms of the plasma mirror, which might be an obvious conclusion known from many other experiments, is not supported by the presented calculations. Moreover, in terms of the much different behavior applying different laser wavelengths (for 800nm distinct sideways peaks turning into a more symmetric structure, for 400nm showing the pronounced forward-backward lobes) there is no understanding provided by the calculations and the interpretation remains unclear. In conclusion, although the measured distributions certainly show an interesting outcome, one needs to provide some understanding in order to have a solid interpretation. For this reason, the manuscript is still not on a level that I would recommend it to be published.

Reviewer #2 (Remarks to the Author):

With this revision, the quality of the manuscript has become much better, both in terms of scientific discussion and readability. I feel that the authors have responded appropriately to the comments of reviewers, including myself. I believe that the analysis using the newly introduced "modified particle-in-cell" method has made the interpretation of the experimental results much clearer. Therefore, I think there is no longer any reason for me to object to publication of this manuscript.

Lastly, I would like to add one thing. The newly introduced method is said to be a "modified particle-in-cell" method, but it is unclear what "modified" means. If this was introduced for the first time in this study, please clearly describe the improvements e.g., in supplementary material. If there is already an example of introduction, it is necessary to add citations.

Reviewer #3 (Remarks to the Author):

The authors have acceptably responded to my previous comments. Adjustments are properly applied. The arrangement of the paper should focus more on clarifying the experimental result in a more timely and complete fashion to help the reader quickly understand the experimental facts. The newly added ultrafast analysis is interesting but arouses more questions. I would only recommend the paper for acceptance after these issues are addressed.

1. How does the intense laser alter the dielectric properties of the nanoparticles? In specific, what kind of parameters are used in the simulation and how is it justified?
2. Does the ultrafast analysis reduce to cases where lower power of illumination is used? In specific, considering 800 nm is well below the gold plasma frequency, why doesn't the intrinsic gold plasma trigger the similar effect?

3. With photons with an energy as high as 400 nm, the impact of the inter-band transition of gold to the ultrafast dielectric property of the nanoparticles can be important. This transition would likely also contribute to a non-linearity of the observed effect. Can the author address the impact of the inter-band and probably also intra-band transition?

Responses to the Referees

Response to Referee 1

The authors have significantly improved the manuscript and taken care of the raised criticism. In particular, a modified particle in cell calculation is presented including the propagation of ions, which in principle should provide a direct comparison to the measured distributions.

We sincerely thank the Referee for his/her recognition of our efforts to improve the work in the last round of review, and his/her advises and comments for further optimization. We have made the corresponding adjustments related to the questions and suggestions. We hope the newly submitted work can provide a clear interpretation of the novel processes.

[1] However, the shown results of the calculation raise questions on the interpretation and do not provide a conclusive understanding of the process. The distributions of protons (Fig. 4 a-e) show a preferred direction along the laser polarization axis (sideways with respect to the laser propagation) which even in panel d and e still has minima in the forward and backward direction. One would expect this preference along the laser polarization as also the electric field enhancement (Fig 4 f-i) clearly shows this behavior. On the other hand, the calculated far field enhancement is surprisingly strongly forward-backward peaked. I would expect that from the simulations one should get an idea what is the main mechanism to do this.

We thank the Referee very much for this comment. It is a major discovery presented in this manuscript and should be clarified with solid interpretations.

The PIC calculation results show that the near-field distributions of ions in the real space locates sideways with respect to the laser propagation. This is mainly determined by the induced near field distribution. The near-field feature applies for

the cases at both 400 nm and 800 nm excitations. However, we need to keep in mind that the scalar space data (the real-space distributions) cannot be directly mapped to the vector space data (the momentum-space distributions). The initially produced ions around the nanosphere will propagate to the far field until they reach a steady state in the momentum space. The final momentum distributions are dominated by the forces acting on each particle, especially the Coulomb force acting on each particle initially. The Coulomb repulsive forces acting on the flying protons are from the heavier ions including C^+ , O^+ , Na^+ , Au^+ , CH_n^+ etc. which have similar initial near-field distributions compared to protons. And due to the fact that protons fly much faster than the other heavier ions, the effective Coulomb force can be approximated to be from a fixed positive charge locating close to the nanoparticle surface. Moreover, due to the shielding effect of metal materials, the initial Coulomb force acting on the protons, which influence the most significantly in their propagations, are mainly exerted by the effective charge locates at the same side. The effective charge on the other side could be isolated by the shielded nanosphere. Our far-field momentum distributions are calculated based on this approximation. Since it requires a time period in the range of tens to hundreds of picoseconds to reach the steady state distribution in the momentum space, we cannot run a calculation for such a long time propagation using the PIC model. That might cost infinite calculation time. Here we made a classical propagation calculation for the far-field distributions based on the effective charge approximation, and a reasonable agreement can be obtained between calculation results and experimental observations. **We found that the far-field momentum distribution difference between the cases at 400 and 800 nm can be attributed to the initial asymmetry of charge distributions.**

In Fig. R1, we present the calculated near-field real-space and far-field momentum distributions for protons at the low (a) and the high (b) laser intensities for 400 nm. The enhanced near-field distributions switch from the forward focusing feature to the backward enhanced feature as the laser intensity is increased. Correspondingly, the near-field localization of protons and other heavier ions exhibit similar profiles. One common property of both the field and the ion distributions is that their near-field distributions are **localized** either slightly forward or slightly

backward, which can be clearly recognized in Figs. R1 (a) and (b). In the real experiment there are still difficulties to measure the near-field distributions of various heavier ions, e.g. CH_n^{n+} , C^{n+} , O^{n+} , Na^{n+} , Au^{n+} . However, the Coulomb forces exerted by all these heavy ions can be replaced by an effective force induced from a frozen charge. This is the key approximation in our work for obtaining the far-field momentum distributions of protons based on the classical Coulomb model. The generated protons with a certain near field distribution will be propagated under the effective Coulomb force until they reach a stable state in the momentum space. For the weak laser excitation at 400 nm wavelength, as shown in Fig. R1 (a), the effective charges locate slightly in the forward direction along the laser propagation direction. In Fig. R1(b), as the excitation intensity dramatically increased, the effective charge locates slightly in the backward direction along the laser propagation direction. Due to the displacement of the effective force from the symmetry position, the initial forces acting on most of the protons are along the y axis, i.e. parallel to the laser propagation direction. There will be barely protons with $p_y=0$ in the final state, i.e. the minima along the p_y axis as displayed in Figs. R1(c) and (d). The calculated far-field momentum distributions thus show a main population along the propagation direction, i.e. forward and backward distributions shown in Figs. R1 (c) and (d). The initial asymmetry of the charge distribution is the main reason to cause the forward and backward preferred final momentum distributions of protons at 400 nm excitations.

For the cases at 800 nm excitations, the observed proton momentum distributions show distinct sideways peaks turning into a more uniform structure. The main difference is the initial charge distributions caused by the enhanced near field distribution. As shown in Fig. R2, the calculated near-field real-space and far-field momentum distributions for protons are presented at the low (a) and the high (b) laser intensities for 800 nm. **It is obvious that the initial distributions of protons in real space are more symmetric along the laser propagation direction at 800 nm excitation** compared with the cases in 400 nm. The effective charges of the heavier ions are marked by the circles. **Their displacement with respect to $y=0$ is much smaller compared to that in the 400 nm cases. As a result, this tiny initial asymmetry of the charge distributions will cause a more symmetric ion**

momentum distributions in the final state, where there exist some protons with $p_y=0$ as displayed in Figs. R2 (c) and (d). The final momentum distributions for 800 nm show no pronounced forward and backward lobes which occur for the 400 nm excitation.

Fig. R1. Near-field real-space proton distributions for weak (a) and strong (b) driving lasers at 400 nm excitation wavelength. The laser propagates to the +y direction. The effective charges of heavier ions are marked by the circles. (c) and (d) are the corresponding calculated far-field momentum distributions for protons.

Fig. R2 Near-field real-space proton distributions for weak (a) and strong (b) driving lasers at 800 nm excitation wavelength. The laser propagates to the +y direction. The effective charges of

heavier ions are marked by the circles. (c) and (d) are the corresponding calculated far-field momentum distributions for protons.

This behavior caused by the initial asymmetry has also been demonstrated by the experimental investigations using Reaction Nanoscopy, where the mapping relation between the initial ion yield landscape on the nanoparticle surface and the final momentum distributions can be extracted [Nat. Commun. 10, 4655 (2019), Optica 9, 551 (2022)]. It has been pointed out in Optica 9, 551 (2022) that “For the protons born near the hot spots, the radial emission prevails because of the high symmetry of the local charge distribution, leading to a one to one mapping between birth and final angles. For the protons created at the surface positions away from the hot spot, due to the additional Coulomb force from the higher local charge at the hot spots, the actual final emission direction will be deflected to a larger angle as compared to the angle expected for radial emission alone.” This demonstrates that **the degree of symmetry of the initial charge distribution has a large influence on the final momentum distributions.**

On the other hand, from further calculations of the proton emission for both 400 and 800 nm excitations, we figured out a different intensity-dependent feature in the far-field momentum distributions. As shown in Fig. R3, the calculated near-field and far-field distribution at 400 nm switched from the forward to the backward enhancement within a very small intensity range, such as shown in Figs. R3 (b, c) and the corresponding far-field distributions in Figs. R3 (g, h). The red values indicated in the figures are the intensities in unit of TW/cm^2 . The forward to backward distribution switches within a range of $20 \text{ TW}/\text{cm}^2$. However, the case is different for the 800 nm excitations. As is shown in Figs. R3 (k, m) and (p, r), a slightly forward to backward change can be recognized when the intensity increases from ~ 30 to $130 \text{ TW}/\text{cm}^2$. The switching occurs in a much wider intensity range. This feature also contributes to a direct forward and backward enhancement switch observed for the 400 nm excitations, and a gradually change from forward to sideward, and to backward for the 800 nm excitations, as the laser intensity is increased.

Figure. R3 Intensity dependence of the near-field and far-field distributions at 400 and 800 nm, respectively. (a-e) are the intensity-dependent near-field real-space proton distributions at 400nm. The unit for the value indicated in the upper right corner of each image is TW/cm^2 . (f-j) are the corresponding calculated far-field momentum distributions. (k-t) are similar results obtained at 800 nm.

In the revised manuscript, we have included detailed interpretations regarding this comment in Section V of the SI. And have added the following paragraphs in the main text for explanation.

At the end of Page 5 the following sentences have been included.

“ As confirmed by previous work²³ the radial emission dominates for the protons born near the hot spots because of the high symmetry of the initial local charge distributions. In this case, a one-to-one mapping between the birth and the final emission angles can be obtained. However, for a highly asymmetric local charge distribution generated at 400 nm excitation, the actual final emission direction of protons will be deflected to a larger angle. A slight deviation in the degree of asymmetry in the near-field distribution can cause a dramatic difference in the final momentum distributions. The 400 nm excitation can cause a clear forward focusing of the H^+ ions at low excitation intensity. As shown in Fig. 2a, the measured momentum

distributions of protons emitted from nanoparticle exhibits a clear forward focusing pattern for the 400 nm excitation at laser intensities below 10 TWcm^{-2} . While for the 800 nm excitation, the protons exhibit a forward focused two-lobe distribution along the polarization direction.”

At the end of Page 8 the following sentences have been included.

“The far-field momentum distribution of the H^+ ions can be obtained from a proceeding calculated based on the classical model (details can be found in Section I of the SI), and the corresponding results are shown in Fig. 4j. Here, what we need to pay attention is that, the real-space distributions could be quite different from the momentum distributions for ions. The far-field momentum distribution is a propagation result of the initially localized protons under the action of Coulomb forces from the other heavier ions. The degree of asymmetry of the initial charge distributions can result in dramatically different final momentum distributions. Radial emission is more probable for protons born with higher symmetry of the local charge distributions²³, leading to a one to one mapping between the birth and final emission angles. In this condition, the far-field momentum distribution can inherit the main features of the near field distribution. However, with an asymmetric initial charge distribution, the protons would be preferred to be reflected towards larger angles, resulting in the forward and backward distribution observed at 400 nm. When the driving laser wavelength is 800 nm, the near-field enhancement is more delocalized and more symmetric with respect to the laser polarization direction compared to the cases at 400 nm. This difference leads to different far-field momentum distributions for 400 nm and 800 nm excitations.”

[2] Furthermore, there are no calculated results for different intensities to be compared with the measured distributions, indicating the right behavior of changing the emission direction. In contrast, it is not clear at all why the calculated distribution is so much peaked in the forward-backward direction, and how to understand the results for the different laser conditions.

We thank the Referee very much for pointing this out. This is indeed a main discovery of our work and should be interpreted clearly. We have made further calculations for the near-field real-space (Figs. R3(a-e) and (f-j)) and the corresponding far-field momentum distributions (Figs. R3(k-o) and (p-t)) of protons at different laser peak intensities at 400 nm and 800 nm, respectively. At both 400 nm and 800 nm, the dominant far-field momentum distributions switch from forward to backward as the laser intensity increases, which is consistent with our experimental data. The momentum distributions for 400 nm driving laser shows a minimum along the polarization direction of the laser field at all the investigated intensities. While, for the 800 nm excitations, the momentum distributions do not show such a minimum along the polarization direction. This is mainly attributed to the initial asymmetry of the real-space near-field cation distributions. This phenomenon has been explained in Ref. [All-optical nanoscopic spatial control of molecular reaction yields on nanoparticles, *Optica* 9, 551 (2022).] as “For the protons born near the hot spots, the radial emission prevails because of the high symmetry of the local charge distribution, leading to a one-to-one mapping between birth (real space position) and final angles (momentum vector direction)”. This means that a high symmetry in the initial charge distributions can cause fewer changes in the final distributions along the laser propagation direction. While for the cases at 400 nm, the born ions exhibit a higher asymmetry thus the asymmetry in the final momentum distribution is amplified. The main difference for the two excitation wavelengths has been discussed in detail in the response of Question [1].

As been indicated is the response of Question 1, we have added the calculation results for different intensities in Section V of the SI, an extra paragraph in the main text for explanation.

[3] In the end, the interpretation in terms of the plasma mirror, which might be an obvious conclusion known from many other experiments, is not supported by the presented calculations.

We thank the Referee for the very careful reading and for raising this question. In

the analysis of the last round we figured out that the interpretation of “plasma mirror” may not be appropriate due to the fact that no real reflection can be observed in our calculation results in this sub-wavelength spatial scales. However, if we test at a macroscopic region with the same local charge densities, reflection of a propagating optical wave can be obtained in the calculation results. Nevertheless, the “plasma mirror” effect may not be a proper term for nanosystems. We have changed the wordings to “resonant absorption induced ionization” where a critical electron density is required, similar to the macroscopic “plasma mirror” effect. The induced near field enhancement at the backside of the nanoparticle caused a dramatic enhancement of absorption and the resulting ionization and dissociation process generate quite a lot charged particles and the local charge density can achieve the critical density.

This has been clarified in the last paragraph on Page 9 as follows.

“In this strong field regime, the free electron density created at the leading edge of the laser pulse can quickly reach a critical value at the backside of the nanoparticle, where dramatic resonance absorption occurs. Considerable ionizations can be induced locally and result in photoion ejections towards the backward directions. Briefly, the fundamental mechanism of process can be described by the following steps: 1. Forward emission: the external field strength is relatively low at the leading edge of laser pulse (lasting for a few optical cycles), the near-field enhancement of gold nanoparticles obeys the Mie theory, and protons are mainly emitted in the forward direction. 2. Electron accumulation: The laser field keeps growing with time, rapid aggregation of free electrons will occur within femtosecond time scales at the backside of the nanoparticle. 3. Resonant absorption induced ionization: the density of the electrons can reach a critical value and resonant absorption takes place at the backside of the nanosystem^{15,21}.”

[4] Moreover, in terms of the much different behavior applying different laser wavelengths (for 800nm distinct sideways peaks turning into a more symmetric structure, for 400nm showing the pronounced forward-backward lobes) there is no understanding provided by the calculations and the interpretation remains unclear.

We thank the Referee very much for pointing this out. We believe this question has been clarified in the response of Question [1].

In conclusion, although the measured distributions certainly show an interesting outcome, one needs to provide some understanding in order to have a solid interpretation. For this reason, the manuscript is still not on a level that I would recommend it to be published.

We thank the Referee so much for these valuable questions and comments, which indeed help us to optimize the interpretation of our work. Further calculations have been performed accordingly and the related results have been included in the revised version. We hope that we can clarify the key points in the newly submitted manuscript.

Response to Referee 2

With this revision, the quality of the manuscript has become much better, both in terms of scientific discussion and readability. I feel that the authors have responded appropriately to the comments of reviewers, including myself. I believe that the analysis using the newly introduced "modified particle-in-cell" method has made the interpretation of the experimental results much clearer. Therefore, I think there is no longer any reason for me to object to publication of this manuscript.

We appreciate the Referee for his/her recognition of our effort to improve the manuscript.

Lastly, I would like to add one thing. The newly introduced method is said to be a "modified particle-in-cell" method, but it is unclear what "modified" means. If this was introduced for the first time in this study, please clearly describe the improvements e.g., in supplementary material. If there is already an example of introduction, it is necessary to add citations.

We thank the Referee very much for this question. The “modified particle-in-cell” method used in this work is actually a combined theoretical calculation of PIC and classical Coulomb model. To obtain a reasonable understanding of the underlying dynamics of the interactions between femtosecond laser fields and gold nanoparticles, the calculations need to consider the complex ionization processes and the succeeding particle propagations to a time scale of tens of picoseconds, which is the time scale required for the multi-particle system to reach a far-field stable state. As an *ab-initio* calculation method, the PIC numerical simulation is extremely time-consuming. It is unacceptable to propagate all the particles to a few tens of picoseconds. Thus, in the so-called **modified** PIC method, we firstly calculate the dynamics based on the general PIC method to a proper time scale, e.g. 100 fs, to get a real-space near-field distributions for all particles. Then, the far-field momentum distributions are obtained

by propagating certain particles using a classical Coulomb exclusion model. The Coulomb repulsive forces acting on the flying protons are from the heavier ions including C^+ , O^+ , Au^+ , CH_n^+ etc. which have similar initial near-field distributions compared to protons. And due to the fact that protons fly much faster than the other heavier ions, the effective Coulomb force can be approximated to be from a fixed point charge locating close to the nanoparticle surface, as shown in Figs. R1 (a,b) and R2 (a,b).

To clarify this, we have added Reference [26] for the PIC method on Page 8 and pointed out at the end of this page that “The far-field momentum distribution of the H^+ ions can be obtained from a proceeding calculated based on the classical model (details can be found in Section I of the SI)... ” The related details have been included as PART V in the Supplementary Information.

Response to Referee 3

The authors have acceptably responded to my previous comments. Adjustments are properly applied. The arrangement of the paper should focus more on clarifying the experimental result in a more timely and complete fashion to help the reader quickly understand the experimental facts. The newly added ultrafast analysis is interesting but arouses more questions. I would only recommend the paper for acceptance after these issues are addressed.

We thank the Referee very much for the questions which help us to further improve the work.

[1] How does the intense laser alter the dielectric properties of the nanoparticles? In specific, what kind of parameters are used in the simulation and how is it justified?

Compared to dielectric nanoparticles, metal nanomaterials exhibit different properties when interacting with intense laser fields. In our work, gold nanospheres are used in the experiment. When the laser pulse shines on the nanosphere, the electromagnetic field is shielded inside the gold nanoparticle. The skin depth of the laser is decreased as the laser intensity becomes stronger. As indicated in Ref. [Strong-Field Control of Plasmonic Properties in Core-Shell Nanoparticles, ACS Photonics 9, 3515 (2022)], the skin depth is about 3 nm for a 5 TW driving laser. Ionizations only occur at the surface of gold nanoparticles. As the nanosystem has been sufficiently ionized, a nanoplasma can be formed. At this point, the whole system acts like a core-shell nanosystem. The core is the pure gold nanoparticle and the shell is the induced plasma.

If a dielectric nanoparticle interacts with very intense laser field, complex ionization and scattering processes may induce carrier avalanching in the volume of the particles, resulting in subcycle metallization of nanoparticles [Ionization-Induced

Subcycle Metallization of Nanoparticles in Few-Cycle Pulses, ACS Photonics 7, 3207 (2020)]. The intense laser field and nanoparticle interaction can generate a large amount of charged carriers in a small volume. Upon expansion of the resulting over dense nanoplasma, the plasma density can drop to the value required for establishing resonance between the Mie plasmon and the driven laser field [Mechanisms of cluster ionization in strong laser pulses, J. Phys. B: At., Mol. Opt. Phys. 39, R39 (2006)]. The resulting transient resonant excitation of the Mie plasmon can lead to extreme energy absorption [High Intensity Laser Absorption by Gases of Atomic Clusters, Phys. Rev. Lett. 78, 3121 (1997)]. Due to the severe carrier dynamics in the microscopic scales, the localized dielectric properties can be dramatically modified on subcycle time scale, e.g. an ultrafast switching to metallic phase.

In our PIC simulations, the main parameters include the size of nanoparticle, the atomic density, the atomic ionization energy, wavelength, and laser intensity. The numerical value of the parameters have been included in the manuscript. The diameter of the nanoparticles is consistent with the experiment. The atomic density of gold is given by the density of solid gold. The ionization energy of atoms can be directly given by EPOCH code package after specifying the atomic number. The wavelength and laser intensity are consistent with the parameters used in the experiment.

[2] Does the ultrafast analysis reduce to cases where lower power of illumination is used? In specific, considering 800 nm is well below the gold plasma frequency, why doesn't the intrinsic gold plasma trigger the similar effect?

Thanks for the Referee for this question. According to the Drude model, the gold plasma frequency mentioned here can be considered as a polarization response of the metal material excited by an electromagnetic field. It corresponds to an internal electrostatic oscillations of plasma. The electric field pulls the electrons back towards equilibrium, where they exactly neutralize the ion charge, and the kinetic energy gained in the process causes the electrons to overshoot to a new displacement on the other side. In this process, the electrons are not pulled too far from the ion core. However, in our cases where intense laser fields at 800 or 400 nm are used to excite

the gold nanoparticles (the induced near field can be enhanced to the level of 10^{14} W/cm², orders of magnitude higher than the fields used to excite the intrinsic gold plasma), the molecules attached to the particle surface and a thin depth of Au atoms at the surface of the nanosphere can be ionized, where many of the freed electrons will escape the system in a very short time scale, e.g. in femtoseconds. The intense-field processes are intrinsically different compared to the process here indicated by the Referee. The intrinsic gold plasma mentioned here by the Referee is in a totally different regime. Therefore the gold plasma does not trigger the similar effect reported in our work.

In the strong-field regime, the ultrafast analysis based on modified PIC method can apply to cases at slightly lower power of illuminations, e.g. where the Mie theory can provide reasonable predictions. However, the availability can only be applied to lower intensities around, e.g. 10^{12} W/cm², where ionizations can still be initiated. Without enough charged particle generations, the PIC method does not make sense. This method applied to the strong-field regime.

[3] With photons with an energy as high as 400 nm, the impact of the inter-band transition of gold to the ultrafast dielectric property of the nanoparticles can be important. This transition would likely also contribute to a non-linearity of the observed effect. Can the author address the impact of the inter-band and probably also intra-band transition?

We thank the Referee for this professional comment.

When gold nanoparticles are excited with 400 nm laser pulses, interband and intraband transition can be induced in the system. These processes can have an impact on the properties of localized surface plasmons (LSPs) for gold nanoparticles. The electrons in the d band need at least 2.4 eV to reach the Fermi level [Quantifying plasmon resonance and interband transition contributions in photocatalysis of gold nanoparticle, Chin. Phys. B 30, 077301 (2021)]. The photon energy of 400 nm is about 3.1 eV. Therefore the electron in the d band can be excited to a maximum of 0.7

eV energy above the Fermi level for interband transition. The hot electrons excited by the interband transition can convert the energy into heat through the non-radiative relaxation or transfer to molecules to initiate the chemical reactions [Nat. Nanotechnol. 10, 770 (2015); Nano Lett. 16, 1478 (2016); J. Phys. Chem. Lett. 8, 4925 (2017)]. While for the intraband transition, electrons can be excited on the conduction band and exceed the Fermi level at a higher energy up to 3.1 eV and could participate in the surrounding ionization reactions.

In the present laser intensity regime, the induced near field enhancement can reach up to an intensity of $\sim 10^{14}$ W/cm². According to the Keldysh parameter, which has been widely served as an criteria to the dominated ionization process in the strong-field physics, the electron release in the present work should be dominated by multiphoton or tunneling ionization processes, which has also been demonstrated by former works, e.g. Nature 616, 702 (2023), Phys. Rev. A 94, 051401 (R) (2016). The strong field induced nanoparticle reaction can generate photoelectrons with energies up to $\sim 200 U_p$, where U_p is the ponderomotive energy. Compared to the electrons generated from multiphoton and tunneling ionizations in the nanosystem, the hot electrons induced by interband or intraband transitions are in general with much lower energies and very probably with lower yields. The influence can be negligible. The exact contribution of the interband and intraband transitions is hard to be determined in the present theoretical framework.

Reviewers' comments:

Reviewer #1 (Remarks to the Author):

The authors responded in a relatively long document to address the raised questions. There, with no doubt, they confirm the validity of their interpretation and provide useful information to understand how the simulations are done and how to understand the outcome. However, in the manuscript, there are only two new shorter sections regarding the interpretation; the structure and the figures are not changed. To my view, the manuscript still is not in good shape. E.g., in the new text they point to a comparison of the experimental data (Fig. 2) with simulations shown in the SI. It takes some time for the reader to find out what has to be compared to what. Moreover, the simulation are not shown at the laser powers used in the experiment. With the extensive simulations made, one should have a direct comparison of experiment and simulations in the manuscript to make it easy for the reader to follow the analysis and interpretation. After two rounds of review and extensive text from the authors, I truly believe that they have done sound work, however, for everyone, who has not digested all this information, and read the material several times, the manuscript does not provide a clear-cut presentation of data and interpretation. My conclusion is that the manuscript now should be rejected. It is not the responsibility of the reviewers to make in further discussions the required upgrades.

Reviewer #3 (Remarks to the Author):

The authors have properly addressed my questions and comments. I recommend the paper for acceptance at this stage.

In this response letter, we put the *original comments by the Reviewers in italics* and black to distinguish them from our responses in blue.

Responses to the Referees

Response to Referee 1

1. The authors responded in a relatively long document to address the raised questions. There, with no doubt, they confirm the validity of their interpretation and provide useful information to understand how the simulations are done and how to understand the outcome. However, in the manuscript, there are only two new shorter sections regarding the interpretation; the structure and the figures are not changed.

We appreciate the reviewer's recognition of our effort. Based on the concerns raised by the reviewer, we have made further upgrades in this revision. The data have been reorganized to show a direct comparison between experimental and simulation results in the main text, this can be seen in the renewed Figure 2 and Figure 5 in the main text (which are also shown below). The corresponding interpretation and discussions have been improved in the main text, which can be seen in Sections I and II in the main text. The structure and presentation are in a good condition now.

Figure 2 (a-c) The integrated momentum distributions of protons generated from surface molecular ionization in gold nanosystems at the intensities of 5, 75, and 100 TWcm^{-2} at 400 nm, respectively. The black and the red arrows in (a) indicate the laser propagation and polarization directions, respectively. (d) Polar plots of the proton momentum distributions at 5 TWcm^{-2} and 100 TWcm^{-2} . (e) Calculation results of the near field $|E|$ distributions based on FDTD for 100 nm gold nanospheres interacting with low intensity laser pulses at 400 nm. (f) The calculated far-field momentum distributions of protons ejected from gold nanoparticles irradiated by 400 nm. Inserted is the initial spatial distributions of the protons from gold nanospheres. The plus signs at (-40 nm,-15 nm) and (40 nm,-15 nm) represent the effective charges for the localized plasmas.

Figs. 2e and 2f are moved here from the SI for better comparison of experimental and theoretical calculations.

Figure 5 (a-e) Calculated intensity-dependent near-field real-space proton distributions at 800 nm. The unit for the value indicated in the upper right corner of each image is TWcm^{-2} . (f-j) and (k-o) are the calculated and measured far-field momentum distributions, respectively. (p) The normalized angular distributions of the ion yields for various excitation strengths. The data at various intensities was multiplied by a factor for better illustration. (q) The peak emission angles as a

function of the total ion yields. The red dashed line is obtained by a linear fitting based on the formula indicated in the figure. (r,s) are three-dimensional isosurface of the topological measurement results obtained at two typical laser intensities.

Figs. 5f-o are moved here from the SI for direct comparison of experimental and theoretical calculations at different laser intensities.

The reviewer's comment "*there are only two new shorter sections regarding the interpretation; the structure and the figures are not changed*" is not totally right since we have added new figures in the main text (Figure 4), and in the SI (Figure S9) in the last round of review. Behind these data are the authors' efforts to produce a new theory model and explore novel interaction regime which has never been revealed before. However, we apologize for the presentation of data which might cause inconvenience for the readers, this issue has been addressed in the above modifications.

2. To my view, the manuscript still is not in good shape. E.g., in the new text they point to a comparison of the experimental data (Fig. 2) with simulations shown in the SI. It takes some time for the reader to find out what has to be compared to what. With the extensive simulations made, one should have a direct comparison of experiment and simulations in the manuscript to make it easy for the reader to follow the analysis and interpretation.

We thank the reviewer very much for this point. We apologize for the inconvenience caused for the readers. Due to the length limitation and the large amounts of data to be presented, we have put some simulation results in the SI in the last round. To solve this problem, we have reorganized the data and have put the experimental and simulation results together in the main text for a direct comparison. The renewed Figures 2 and 5 were added in the text and the corresponding interpretations were updated. Renewed figures can be seen in the response of the last comment.

3. Moreover, the simulation are not shown at the laser powers used in the experiment.

We thank the reviewer very much for finding this drawback. Actually, in the previous round, we have provided the calculation results of final-state momentum distributions in Fig. S9 in the SI for a wide range of laser intensities that fully include all the corresponding experimental laser intensities. However, the data presented in

Figure 4 in the main text, which is used to show the underlying dynamics, was presented at 200 TWcm^{-2} , which is higher than the intensities used in our experiment. We are very sorry for the confusion caused by this. We have renewed this figure with updated data calculated at 100 TWcm^{-2} within the experimental intensity range. The new Figure 4 can be seen below. As a matter of fact, the behaviors at 100 and 200 TWcm^{-2} are similar, while the former results at 200 TWcm^{-2} is to show the phenomena under more extreme conditions, where a backward proton ejection is more apparent in the near-field real-space distributions.

Renewed Figure 4 PIC calculation results (EPOCH2D version 4.18.0). (a-e) The instantaneous distributions of protons generated from surface H_2O molecular ionizations in gold nanoparticles ($d = 100 \text{ nm}$) at 25, 30, 35, 40, 80 fs in the real space, respectively. The excitation laser parameters are, center wavelength 400 nm, peak intensity 100 TWcm^{-2} , pulse duration 30 fs (full width at half maximum). The black and the red arrows in (a) indicate the laser propagation and polarization directions, respectively. (f-i) Averaged distributions of the enhanced electric field around the nanoparticle for the period of 24-26 fs, 29-31 fs, 34-36 fs, and 39-41 fs. (j) Calculated far-field momentum distributions for protons. The data was interpolated and smoothed due to the large difference in scale between the initial and final states. Inserted is the corresponding experimental single-shot data.

For a comparison, the previous results in Figure 4 obtained at 200 TWcm^{-2} is presented below.

Figure 4 in the last round, obtained at 200 TWcm^{-2} .

4. After two rounds of review and extensive text from the authors, I truly believe that they have done sound work, however, for everyone, who has not digested all this information, and read the material several times, the manuscript does not provide a clear-cut presentation of data and interpretation. My conclusion is that the manuscript now should be rejected. It is not the responsibility of the reviewers to make in further discussions the required upgrades.

We thank the reviewer very much for pointing out the drawbacks. However, after carefully going through these comments, none of the issue is unsolvable. We have made proper upgrades to make it easier for the readers to see a clear comparison of experimental and simulation results of the extreme interactions between nanoparticles and intense laser fields. All the issues have been addressed. We hope the novelty and significance of this work can be recognized.